# Extreme electron–hole drag and negative mobility in the Dirac plasma of graphene

Leonid A. Ponomarenko [1,2] ✉, Alessandro Principi[2] ✉, Andy D. Niblett[1], Wendong Wang [3], Roman V. Gorbachev [3], Piranavan Kumaravadivel [3], Alexey I. Berdyugin[2], Alexey V. Ermakov [2], Sergey Slizovskiy [2,3], Kenji Watanabe [4], Takashi Taniguchi [4], Qi Ge[5], Vladimir I. Fal'ko [2,3], Laurence Eaves [6], Mark T. Greenaway [6,7] ✉ & Andre K. Geim [2,3] ✉

Coulomb drag between adjacent electron and hole gases has attracted considerable attention, being studied in various two-dimensional systems, including semiconductor and graphene heterostructures. Here we report measurements of electron–hole drag in the Planckian plasma that develops in monolayer graphene in the vicinity of its Dirac point above liquid-nitrogen temperatures. The frequent electron–hole scattering forces minority carriers to move against the applied electric field due to the drag induced by majority carriers. This unidirectional transport of electrons and holes results in nominally negative mobility for the minority carriers. The electron–hole drag is found to be strongest near room temperature, despite being notably affected by phonon scattering. Our findings provide better understanding of the transport properties of charge-neutral graphene, reveal limits on its hydrodynamic description, and also offer insight into quantum-critical systems in general.

If electron- and hole-doped two-dimensional (2D) conductors are placed in close proximity to each other, Coulomb interactions between charge carriers in adjacent layers lead to electron–hole drag (for review, see refs. [1,2]). The drag was extensively studied using various electronic systems based on GaAs heterostructures and, more recently, graphene[1–12]. The strength of Coulomb interaction rapidly increases with decreasing the distance between 2D systems, and the ultimately strong drag is expected if electrons and holes coexist within the same atomic plane. Graphene near its Dirac or neutrality point (NP) provides the realization of such an electronic system. Indeed, close to the NP, a finite temperature $T$ leads to thermal excitations of electrons and holes, whereas their relative concentrations can be controlled by gate voltage. The resulting electron–hole plasma is strongly interacting and represents a quantum critical system where particle–particle collisions are governed by Planckian dissipation[13–23]. The system is also often referred to as Dirac fluid, assuming inter–carrier scattering dominates other scattering mechanisms. Because the Dirac plasma in graphene is a relatively simple and tunable electronic system, its behavior can be insightful for understanding of electron transport in more complex Planckian systems including "strange metals" and high-temperature superconductors in the normal state[24,25]. There is also an interesting conceptual overlap with relativistic electron–positron plasmas generated in cosmic events, which are difficult to recreate in laboratory experiments[26]. Previous experimental studies of the Dirac plasma reported its hydrodynamic flow[20], the violation of the Wiedemann–Franz law[17], giant linear magnetoresistance[23], and other anomalies indicative of the quantum-critical regime[18–23]. So far, the possibility of probing mutual drag between electron and hole subsystems within the Dirac plasma has escaped attention.

Department of Physics, University of Lancaster, Lancaster, UK. Department of Physics and Astronomy, University of Manchester, Manchester, UK. National Graphene Institute, University of Manchester, Manchester, UK. Research Center for Electronic and Optical Materials, National Institute for Materials Science, Tsukuba, Japan. Institute for Functional Intelligent Materials, National University of Singapore, Singapore, Singapore. School of Physics and Astronomy, University of Nottingham, Nottingham, UK. Department of Physics, Loughborough University, Loughborough, UK. ✉ e-mail: l.ponomarenko@lancaster.ac.uk; alessandro.principi@manchester.ac.uk; m.t.greenaway@lboro.ac.uk; geim@manchester.ac.uk

# Results

## Longitudinal and Hall resistivity of the Dirac plasma

Our devices were multi-terminal Hall bars made from encapsulated monolayer graphene. It was essential to make them larger than 10 µm in width to avoid an obscuring contribution from edge scattering and charge accumulation at boundaries[12,27]. The devices exhibited high carrier mobilities (~$10^6$ cm² V⁻¹ s⁻¹) and little inhomogeneity (Methods). We studied several such devices, 3 of which were chosen for detailed analysis of their longitudinal and Hall resistivities near the NP ($\rho$ and $R_H$, respectively). All of them showed practically identical characteristics so that, for brevity and consistency, we illustrate the observed behavior using the data obtained from one of the devices. Its optical micrograph is shown in the inset of Fig. 1a.

Near the NP, where both electrons and holes are present, the total charge density in graphene is given by $en = e(n_e − n_h)$ where $n_e$ and $n_h$ are the sheet densities of electrons and holes, respectively, and $e$ is the electron charge. The charge density $en$ can be controlled capacitively by gate voltage (Supplementary Note 1). The device's resistivity $\rho$ as a function of $n$ is shown in Fig. 1a (positive and negative $n$ correspond to electrons and holes, respectively). At low $T$, $\rho(n)$ exhibits a sharp peak at the NP (red curve). It is instructive[18,23] to replot $\rho(n)$ in a logarithmic scale (right inset) which reveals that $\rho$ is weakly density-dependent for $n \lesssim 10^{10}$ cm⁻². The point at which $\rho$ becomes notably dependent on $n$ is labeled as $\delta n$ (arrow in the inset of Fig. 1a). The value of $\delta n$ at low $T$ provides a measure of residual charge inhomogeneity ("electron–hole puddles")[18,23]. Despite its extra-large size (15 × 30 µm²), the device exhibited $\delta n$ of only ~$5 \times 10^9$ cm⁻² at low $T$. As $T$ increased, the peak in $\rho(n)$ became wider and smaller because of thermally excited electrons and holes (black curves in Fig. 1a, b). At room $T$, the measured value of $\delta n$ increased by an order of magnitude with respect to that at liquid–helium $T$ (inset of Fig. 1b).

The corresponding behavior of Hall resistivity $R_H$ near the NP is shown in Fig. 1c. A small magnetic field $B$ was applied perpendicular to graphene, and its value (4 mT) was carefully chosen to keep electron transport deep in the weak–field limit where $R_H$ remained linear in $B$ (nonlinearities started emerging typically above 10 mT) and, at the same time, to ensure a large enough Hall response to record $R_H$ with high accuracy. Both conditions were essential for our analysis described below. Away from the NP, at densities $|n| > 10^{11}$ cm⁻², $R_H$ evolved as $B/ne$, as expected for transport dominated by one type of charge carriers (Fig. 1c). Near the NP, $R_H(n)$ departed from this dependence due to the presence of both electrons and holes and changed its sign at the NP, indicating a switch from majority hole to majority electron

transport. The range of $n$ over which both electrons and holes contributed to the Hall effect can be characterized by $\delta n_H$, the distance between the maximum and minimum in $R_H$ (see the inset of Fig. 1c). $\delta n_H$ did not depend on $B$ (in the discussed limit of weak $B$) and increased with $T$ as the density of thermally excited charge carriers increased. This is illustrated in Fig. 2a which shows $R_H(n)$ at three different $T$. As the temperature increased, the extrema in $R_H$ were broadened and moved further apart. Figure 2b shows $\delta n_H$ measured over a wide range of $T$ and compares the behavior with $\delta n(T)$ determined from the broadening of the peak in $\rho(n)$. Both $\delta n_H$ and $\delta n$ exhibit similar values and a roughly parabolic $T$ dependence. At low $T$, they tend to have a constant value due to residual charge inhomogeneity.

The transport behavior described above and illustrated by Figs. 1 and 2 is archetypical of high-quality graphene. It was previously observed in numerous experiments but not subjected to in-depth analysis. Most often, $\rho(n)$ curves have been used only to evaluate the charge inhomogeneity of a device (as described above) and extract the field-effect mobility defined as $\mu(n) = 1/ne\rho(n)$. The latter expression is valid only in the case of one type of carrier so that, unsurprisingly, $\mu$ has been found to diverge near the NP because $n$ goes through zero (blue curve in Fig. 3a). As for the behavior of $R_H(n)$, the region close to the NP has usually been ignored with reference to the presence of electron–hole puddles. This is justified at liquid–He temperatures but, as thermal excitations overpower the effects of charge inhomogeneity with increasing $T$, electron transport at the NP becomes intrinsic. This high-$T$ regime was overlooked previously and merits a better understanding, which is provided below.

## Two-fluid model for the Dirac plasma

For two types of charge carriers present in graphene near its NP, it is sensible to try to describe the transport characteristics using the standard two-carrier Drude model[28]:

$$\rho(n) = \frac{1}{e(\mu_e n_e + \mu_h n_h)} \tag{1}$$

$$R_H(n) = \frac{B}{e} \frac{n_e \mu_e^2 - n_h \mu_h^2}{(n_h \mu_h + n_e \mu_e)^2} \tag{2}$$

where $\mu_e$ and $\mu_h$ are the mobilities of electrons and holes, respectively. Their densities are given by $n_{e,h} = \int f(\pm \varepsilon_k, \Theta) DoS(\varepsilon_k) d\varepsilon_k$ where $f(\pm \varepsilon_k, \Theta)$ is the Fermi–Dirac distribution for electrons (+) and

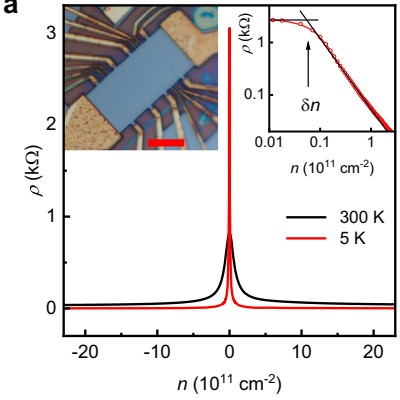
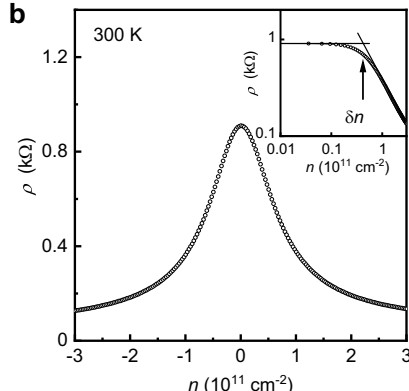
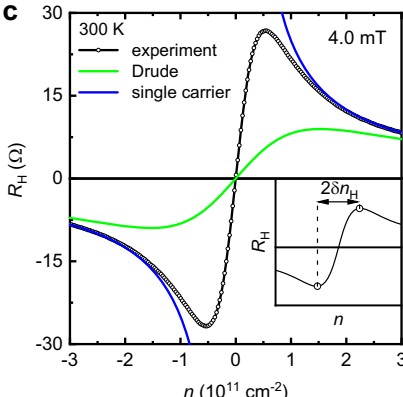

**Fig. 1 | Transport characteristics of monolayer graphene near the neutrality point. a** Resistivity at room and low $T$ in zero magnetic field. Left inset: optical micrograph of one of the studied devices. Scale bar, 10 µm. Right inset: $\rho(n)$ measured at 5 K is replotted on a log–log scale. The arrow marks $\delta n$ at which point the resistivity starts responding to gate voltage. **b** Zooming in on the behavior of $\rho$ in the vicinity of the NP at 300 K (black symbols, same curve as in **a**). Inset:

same as the inset in (**a**) but at room $T$. **c** Room-$T$ Hall resistivity in small $B$ (open symbols). The green curve plots $R_H$ expected from the standard Drude model assuming electron-hole symmetry $\mu_e(n) = \mu_h(n)$ (the curve does not depend on the mobilities' absolute values). Blue curves: $R_H = B/ne$ as expected for a single carrier type. The inset explains how we define $\delta n_H$ that is analogous to $\delta n$ in panels **a, b**.

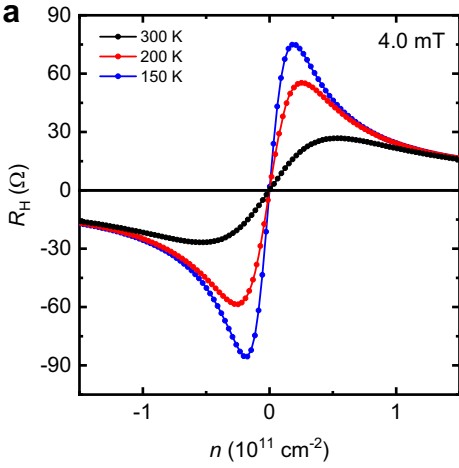
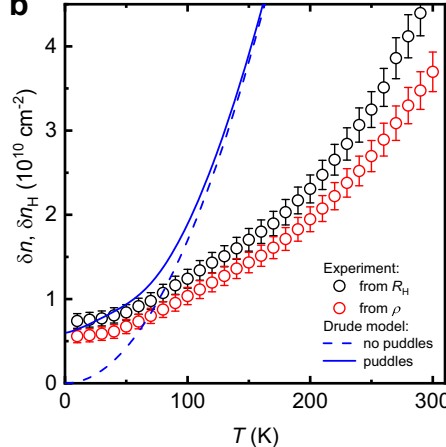

**Fig. 2 | Hall resistivity in low magnetic fields and thermal broadening at the NP.** **a** Examples of $R_H(n)$ at different $T$ (color coded). **b** Characteristic width of the region where both electrons and holes are present. Symbols: $\delta n_H$ and $\delta n$ extracted from the Hall and longitudinal resistivities, respectively. Error bars: standard deviations. Blue curves: $\delta n_H(T)$ expected from the standard Drude model that ignores electron−hole drag.

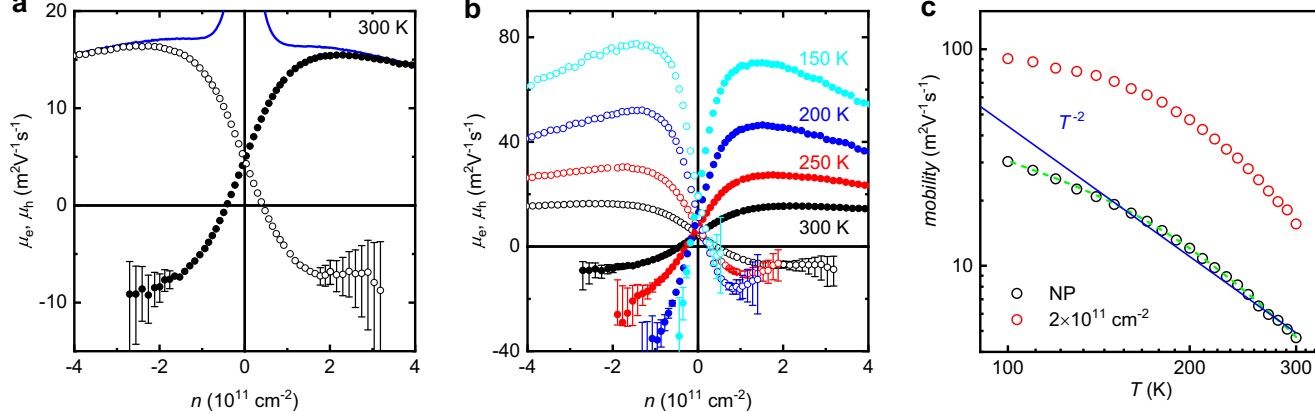

**Fig. 3 | Trade-off between electron and hole mobilities near the Dirac point.** **a** Room−$T$ mobilities of electrons (solid symbols) and holes (open) extracted from Hall and longitudinal resistivities at low $B$ using the modified Drude model. As the density of majority carriers increases away from the NP, their mobility also increases but the mobility of minority carriers rapidly becomes negative. Blue curve: field-effect mobility extracted under the assumption of one type of charge carrier. The error bars arise from noise of ~0.2 Ohm in the measured Hall resistance. **b** Same as in (**a**) but at a different $T$ (color-coded). **c** Mobility at the NP (black symbols) and at the density of $2 \times 10^{11}$ cm$^{-2}$ (red) as a function of $T$. For self-consistency check, the green curve shows the mobility calculated directly from the minimum conductivity rather than using Eq. 3. Blue curve, $T^{-2}$ dependence.

holes (−), and $DoS(\varepsilon_k)$ is the density of states. For a given $n$, the electrochemical potential $\Theta$ can be found by solving the integral equation $n = n_e - n_h$ (Supplementary Note 2), which in turn allows us to find $n_{e,h}$ as a function of $n$. At the NP ($n$ and $\Theta = 0$), $n_e = n_h \equiv n_{th} = (2\pi^3/3)(k_B T/h v_F)^2$ where $k_B$ and $h$ are the Boltzmann and Planck constants, respectively, and $v_F$ is graphene's Fermi velocity. At room $T$, the observed intrinsic broadening $\delta n_H \approx \delta n$ was approximately twice smaller than $n_{th}$ (Fig. 2b). The room-$T$ resistivity of ~0.9 kOhm at the NP (Fig. 1) corresponds to $\mu_e = \mu_h \approx 47,000$ cm$^2$ V$^{-1}$ s$^{-1}$ and yields the scattering rate of ~0.3 ps, in agreement with the Planckian frequency $\tau_P^{-1} \approx C k_B T/h$ where $C$ is the constant of about unity[13–23].

If the electron and hole subsystems were to respond independently to the electric field $E$, as the standard Drude model assumes, the electron−hole symmetry of graphene's spectrum would imply equal drift velocities and, therefore, $\mu_e = \mu_h$ (although the mobilities' value may depend on $n$). Then, Eq. 2 simplifies to $R_H(n) = nB/e(n_e + n_h)^2$ which is independent of scattering times. This dependence is shown in Fig. 1c

by the green curve that has no adjustable parameters. This curve is profoundly different from those observed experimentally. The extrema of the Drude curve are much shallower and occur further away from the NP than in the experiment. It also yields $\delta n_H \approx 2.07 n_{th}$ (dashed curve in Fig. 2b), which is ~4 times larger than $\delta n_H$ measured at room $T$. If a finite charge inhomogeneity is included within the Drude model (solid blue curve in Fig. 2b and Supplementary Note 4), we achieve a better match between experimental and theoretical curves for $\delta n_H$ at low $T$ but obviously, this cannot resolve the discrepancy at high $T$. The failure to explain the sharp transition in $R_H(n)$ near the NP shows that the standard Drude model, assuming non-interacting fluids, is inadequate to describe the Dirac plasma's transport properties.

Next, we relax assumptions and, empirically, allow electron and hole mobilities to be unequal and even negative (the latter contradicts the Drude model's assumptions). Equations 1 and 2 contain two unknown functions $\mu_e(n)$ and $\mu_h(n)$ and, for each $n$, their values can uniquely be evaluated from the two measured variables, $\rho$ and $R_H$.

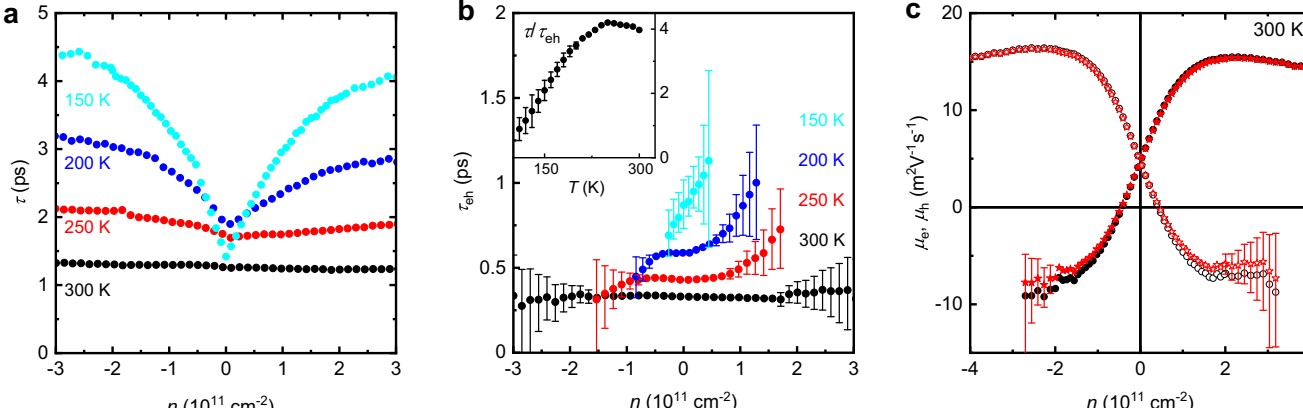

**Fig. 4 | Scattering times in graphene's Dirac plasma and comparison between the Boltzmann and modified Dirac models. a** Extracted $\tau(n)$ caused by impurities and phonons at different $T$ (color coded). **b** Similarly for electron–hole scattering. Experimental errors rapidly increase away from the NP because a contribution of $\tau_{eh}$ towards the transport coefficients is exponentially small beyond a few $n_{th}$. The inset shows the temperature dependence of $\tau/\tau_{eh}$ at the NP. **c** Comparison of the mobilities found using the modified Drude model (black curves, same as in Fig. 3a) and calculated from the Boltzmann model (red) using the scattering times from panels (**a**, **b**).

Combining Eqs. 1 and 2, we obtain the following expression:

$$\mu_{e,h}(n) = \pm \frac{1}{ne\rho}\left[1 - \sqrt{\frac{n_{h,e}}{n_{e,h}}\left(1 - \frac{eR_H}{B}n\right)}\right] \quad (3)$$

The electron and hole mobilities extracted using Eq. 3 are plotted in Fig. 3a, b where we limit our analysis to $T \geq 150\,\text{K}$ so that the density of thermally excited carriers dominates over the residual charge inhomogeneity (Fig. 2b).

At the NP, the electron and hole mobilities are found to be equal as required by symmetry. Away from the NP, as the carrier density of either electrons or holes is increased, their mobility also increases, until it saturates at $n$ of several $n_{th}$, where the charge density is dominated by one type of carrier (see Supplementary Fig. 2). In contrast, the mobility of the minority carriers rapidly decreases away from the NP and becomes negative at $|n| \gtrsim n_{th}$. Near room $T$, the mobility of minority carriers saturates to an absolute value comparable to that of majority carriers (Fig. 3a). This means that the minority carriers are dragged by majority carriers in the direction opposite to their expected drift direction and, if one type of carrier dominates, the other one is forced to drift along with a similar velocity. We observe this reversal in the drift direction for minority carriers over our entire temperature range (Fig. 3b) and for all devices. The behavior is attributed to strong Coulomb interaction between electrons and holes. For completeness, Fig. 3c shows the $T$ dependence of the extracted mobilities at the NP where $\mu_e \equiv \mu_h$ and at a finite density where one carrier type remains present. For charge-neutral graphene, the mobilities evolve approximately as $\propto 1/T^2$ and start saturating below 150 K (Fig. 3c) where electron–hole puddles can no longer be neglected. Square dependence is expected because the Planckian scattering time $\tau_P$ and the effective mass of the Dirac fermions are both linearly dependent on $1/T$ (ref. 29).

## Discussion
### Boltzmann model for the Dirac plasma
Equations 1–3 are inadequate to accurately describe an interacting plasma. The fundamental reason for this is that electron–hole scattering leads to momentum relaxation in the electric field direction but not in the perpendicular Hall field direction[23]. Because of this relaxation anisotropy, electron–hole scattering (described by time $\tau_{eh}$) contributes to the transport coefficients in a different way compared to scattering by phonons and impurities which can be parametrized by another time $\tau$. Accordingly, to describe electron transport in the Dirac

plasma at finite $B$, we have used the linearized Boltzmann model[30] that is presented in Supplementary Note 3. In brief, the Boltzmann model yields the following coupled Drude-like equations:

$$\pm \frac{eE}{m_{e,h}} + \frac{u_{e,h}}{\tau} \pm \frac{\rho_{h,e}}{\rho_e + \rho_h}\left(\frac{u_e - u_h}{\tau_{eh}}\right) = 0 \quad (4)$$

where $u_{e,h}$ are the drift velocities of electrons and holes, $m_{e,h}$ are their energy-dependent effective masses, and $\rho_{e,h}$ are the mass densities (Supplementary Note 3). Both $\rho_{e,h}$ and $m_{e,h}$ are positive and depend on $n$ and $T$ (Supplementary Fig. 3). The first two terms of Eq. 4 have exactly the same form as the standard Drude equation describing the force acting on a charge carrier due to the electric field and an opposing "frictional" force proportional to $1/\tau$. The third term corresponds to an additional frictional force caused by electron–hole scattering. This term can attain a value opposite to the electric field term and dominate over it. In the latter case, charge carriers would be dragged in the direction opposite to $E$. Solving Eq. 4, we determine $\rho$ and $R_H$ as a function of $n$ and the two scattering times (these bulky but analytical expressions are provided in Supplementary Note 3). For each $n$, we again have only two unknowns ($\tau_{eh}$ and $\tau$) that fully define $\rho$ and $R_H$ whereas all the other relevant parameters are determined by graphene's electronic spectrum. The resulting coupled nonlinear equations can be solved numerically, which has allowed us to obtain both scattering times as shown in Fig. 4a, b.

Near room $T$, the extracted $\tau$ is practically independent of $n$. With decreasing $T$, $\tau$ starts exhibiting a dependence close to $\sqrt{n}$ which is expected for charged impurities and other mechanisms sensitive to screening by charge carriers. This square-root dependence yields mobility independent of carrier density[29], typical of graphene at low $T$. At the NP, $\tau$ depends relatively weakly on $T$ over our entire temperature range. Nonetheless, note that $\tau(n=0)$ first increases with increasing $T$, presumably due to stronger screening by the increasingly dense Dirac plasma. Then, above 200 K, $\tau$ decreases because of phonon scattering (Fig. 4a). As for $\tau_{eh}$, it exhibits relatively weak dependence on doping (note that the prefactor in the third term of Eq. 4 accounts for the $n$ dependence of electron–hole friction caused by the varying mass densities). Some electron–hole asymmetry observed below 200 K (Fig. 4b) originates from subtly asymmetric $\rho(n)$ and $R_H(n)$ found in the experiment, probably because of remnant doping. With increasing $T$, $\tau_{eh}$ evolves as Planckian scattering, that is, $\tau_{eh} \approx h/Ck_BT$ where $C \approx 0.6$, in good agreement with the coefficient, reported previously[23]. Furthermore, the inset of Fig. 4b plots the $T$ dependence

of $\tau/\tau_{eh}$ the NP. It exhibits relatively little scatter thanks to the fact that $R_H(n)$ depends only on the ratio $\tau/\tau_{eh}$ rather than the individual times and is very sensitive to its absolute value, which minimizes errors in our numerical analysis (Supplementary Note 3).

We emphasize that the ratio $\tau/\tau_{eh}$ does not exceed 4 at any $T$, meaning that phonon and impurity scattering significantly affect electron–hole drag in the Dirac plasma, especially below 200 K. This bears ramifications for a hydrodynamic description of the Dirac plasma. Indeed, to observe a viscous flow, it is imperative to have particle–particle scattering more frequent than momentum-relaxing scattering. The particle–particle scattering time $\tau_v$ that defines the electron viscosity of the Dirac plasma is generally expected to be comparable to $\tau_{eh}$. This means that, even under the most favorable conditions (close to room $T$), the ratio $\tau/\tau_v$ near the neutrality point is modest (a factor of several at most), suppressing viscous effects, which agrees with recent observations[31,32]. At lower $T$ where values of $\tau_{eh}$ and $\tau$ become close, it would be difficult, if not impossible, to observe even remnants of electron hydrodynamics.

### Justifying the modified Drude model

It is instructive to calculate electron and hole mobilities from the scattering times found using the Boltzmann model (Supplementary Note 3). The results are shown in Fig. 4c for the case of room $T$ where our accuracy was highest because of the largest $\tau/\tau_{eh} \approx 4$. As expected, the Boltzmann analysis also yields negative mobilities for minority carriers at $|n| > n_{th}$ and saturating drift velocities in the same direction for both electrons and holes, if doping is larger than a few $n_{th}$. In the limit $\tau \to \infty$, both electrons and holes are expected to drift with the same velocity (Supplementary Note 3). The finite $\tau/\tau_{eh}$ reduces the drift velocity of minority carriers and, at room $T$, it is approximately twice as small as the velocity of majority carriers. Although the modified Drude model does not distinguish between electron–hole, and electron–phonon scattering, it agrees surprisingly well with the Boltzmann analysis. Notable deviations occur only for minority carriers and do not exceed ~20% (Fig. 4c). The agreement was found to be similar for all the studied devices at $T$ above 150 K. This shows that, however empirical, the Drude model with different and sign-varying $\mu_e(n)$ and $\mu_h(n)$ can be used for a semi-quantitative description of the Dirac plasma in weak fields ($R_H$ should remain linear in $B$; see Supplementary Note 3). Furthermore, both Boltzmann and modified–Drude models describe equally well the measured dependence $\delta n_H(T)$ shown in Fig. 2b (Supplementary Figs. 4 and 5).

### Summary

Graphene's transport characteristics near the NP cannot possibly be understood without considering the strong interaction between the electron and hole subsystems within the Dirac plasma because minority carriers are dragged in the same direction as majority carriers. The observed behavior of both the longitudinal and Hall resistivities is accurately described by our Boltzmann analysis, which allows quantitative evaluation of the scattering rates. Inevitable scattering by phonons and impurities reduces the achievable value of the ratio $\tau/\tau_{eh}$ so that the minority carriers in the Dirac plasma always lag behind the majority ones. For high–quality encapsulated graphene, mutual drag is strongest near room $T$ where the minority carriers drift at approximately half the velocity of the majority carriers. This shows that impurity and phonon scattering significantly affect the transport properties of graphene's Dirac plasma and, in particular, suppresses its viscous (hydrodynamic) behavior.

While the paper was under review, two preprints appeared on arXiv[33,34], reporting magneto and magneto-thermal transport in graphene and interpreting deviations from conventional behavior in terms of electronic viscosity. In all cases, the reported anomalies stem from frequent momentum–conserving electron scattering. However,

unlike our study, the experiments[33,34] required the two-probe Corbino–disk geometry and finite doping and involved significant current-flow distortions caused by viscosity. We used the standard four-probe Hall bar geometry with no detectable distortions in current flow, and we could probe both charge-neutral and doped Dirac plasma, but not the highly degenerate electron liquid as in refs. 33,34. Our analysis allowed direct extraction of mobilities and scattering times ($\tau_{eh}$ and $\tau$) rather than the electron viscosity. A more subtle difference is that the electron-hole drag we observed is caused by electron-hole scattering whereas all momentum-conserving events (described by $\tau_v$) contribute to viscosity in refs. 33,34.

## Methods

The studied devices were made from monolayer graphene encapsulated between two crystals of hexagonal boron nitride. Relatively thick graphite placed under the trilayer heterostructures served as a gate electrode. This allowed charge carrier mobilities to reach ~$10^6$ cm$^2$ V$^{-1}$ s$^{-1}$ at low $T$ (measured at finite carrier densities of a few $10^{11}$ cm$^{-2}$). The remnant doping was low, typically ~$2 \times 10^{10}$ cm$^{-2}$. The electrical measurements were carried out using the standard low-frequency lock-in technique. Further details are provided in Supplementary Information.

## Data availability

The authors declare that the data presented in this study are available on request from LAP.

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

## Acknowledgements

We acknowledge financial support from the European Research Council (grant VANDER) and the Lloyd's Register Foundation (grant Designer Nanomaterials) (A.K.G. and V.I.F.). A.P. was supported by the European Commission under the EU Horizon 2020 MSCA-RISE-2019 program (project 873028 HYDROTRONICS) and the Leverhulme Trust (grant RPG –2023-253). M.T.G. acknowledges support from the Engineering and Physical Sciences Research Council (grant EP/V008110/1). K.W. and T.T. acknowledge support from the Elemental Strategy Initiative of Japan (grant JPMXP0112101001) and JSPS KAKENHI (19H05790, 20H00354, and 21H05233).

## Author contributions

L.A.P. initiated the project, carried out electrical measurements (with help from A.I.B. and A.D.N.), and suggested interpretation using the modified Drude model (with help from A.K.G.). A.P., M.T.G., and A.K.G. provided the analysis using the Boltzmann model. A.K.G., A.P., and M.T.G. wrote the paper with contributions from L.A.P. and L.E. W.W., R.V.G., P.K., K.W., and T.T. provided structures and devices. V.I.F., A.V.E., S.S., A.I.B., and Q.G. provided theoretical support and discussed results.

## Competing interests

The authors declare no competing interests.
