## [Peer Review File · Nature Communications]

REVIEWERS' COMMENTS

Reviewer #1 (Remarks to the Author):

I am satisfied with the authors' responses to the reviewers' comments in the first round and with the corresponding modifications made to the manuscript. I recommend the publication of the manuscript in its current form in Nature Communication.

Reviewer #2 (Remarks to the Author):

I would like to commend the authors for their thorough responses to the comments and questions raised in my initial review. Their efforts in addressing the concerns and clarifying key points have improved the clarity and depth of the manuscript.

In my initial review, I highlighted two concurrent works (arXiv:2406.13799 and arXiv:2407.05026) that utilize the Corbino geometry to explore the hydrodynamic aspects of electronic transport in graphene. The authors have appropriately pointed out that their approach, which is based on the kinetic equation, is distinct from the hydrodynamic description used in these works. While I appreciate the difference in the models employed, I believe some discussion of these concurrent studies is still necessary, particularly because they report transport responses in monolayer graphene under similar ranges of temperature and charge carrier concentrations.

Given the overlap with these concurrent studies, I believe it is crucial for the authors to reference these works in the manuscript. To help readers fully appreciate the models proposed in the current study and differentiate them from the other works, the authors should discuss how the data and models in these papers compare or contrast with their own. If the observed response differs, it would be helpful to understand what differences arise due to sample-specific details and what differences are more fundamental and not accounted for by the existing models. This discussion will enhance the manuscript's relevance and further highlight the importance of the authors' approach.

With these additions, I believe the manuscript will be an even stronger contribution to the field.

Reviewer #3 (Remarks to the Author):

The authors have thoroughly addressed the reviewers' concerns and comments, and I believe the manuscript is suitable for publication in Nature Communications. However, I have one additional question. The observed drag phenomenon near charge neutrality is somewhat reminiscent of ambipolar transport of excited carriers in semiconductors. I would appreciate it if the authors could comment on both the similarities and differences.

Reply to comments of Reviewer #2:

I would like to commend the authors for their thorough responses to the comments and questions raised in my initial review. Their efforts in addressing the concerns and clarifying key points have improved the clarity and depth of the manuscript.

In my initial review, I highlighted two concurrent works (arXiv:2406.13799 and arXiv:2407.05026) that utilize the Corbino geometry to explore the hydrodynamic aspects of electronic transport in graphene. The authors have appropriately pointed out that their approach, which is based on the kinetic equation, is distinct from the hydrodynamic description used in these works. While I appreciate the difference in the models employed, I believe some discussion of these concurrent studies is still necessary, particularly because they report transport responses in monolayer graphene under similar ranges of temperature and charge carrier concentrations.

Given the overlap with these concurrent studies, I believe it is crucial for the authors to reference these works in the manuscript. To help readers fully appreciate the models proposed in the current study and differentiate them from the other works, the authors should discuss how the data and models in these papers compare or contrast with their own. If the observed response differs, it would be helpful to understand what differences arise due to sample-specific details and what differences are more fundamental and not accounted for by the existing models. This discussion will enhance the manuscript's relevance and further highlight the importance of the authors' approach.

With these additions, I believe the manuscript will be an even stronger contribution to the field.

Although the papers present investigations of a similar transport regime in graphene, the focus of the articles are different. Our analysis is based on a simultaneous measurement and self-consistent calculation of the Hall and longitudinal resistance in the Hall bar geometry. This approach allows us to determine that the ratio between the electron-hole scattering rate and the impurity and phonon scattering rate is relatively small. In the highlighted papers, experiments are undertaken in the Corbino geometry where simultaneous longitudinal and transverse measurements are not possible. Therefore, a direct comparison of the measurement/response in these papers with our study and analysis is not straightforward. We also note that arXiv:2406.13799 is focused on thermal transport which is not considered in our approach.

The highlighted articles employ a hydrodynamic description of electron transport to explore the system. For example, in arXiv:2407.05026, the authors' use the Stokes-Ohm model to interpret measurements from the Corbino geometry and separate Ohmic and viscous (electron-electron interactions) contributions. In our analysis, we used a first-principles Boltzmann model which simultaneously accounts for both electron-hole scattering and electron-impurity/phonon scattering. Our analysis revealed that, to explain our measurements, in addition to electron-hole interactions, we cannot disregard the effects of phonon and impurity scattering, especially at low temperature, an approach which is complementary to arXiv:2407.05026. Our conclusion is shared with recent studies (see refs. 31 and 32 of the main text). We note that both arXiv:2407.05026, and our study revealed a linear-in- T dependence of the electron-electron scattering rate.

We feel that extensive comparison of the different experiments would be more appropriate for a review paper and rather than the present report that focuses on our own measurements. Nonetheless, following the Reviewer's comment we have added a new paragraph at the very

end of the revised manuscript that acknowledge the arXiv papers and briefly discusses the mentioned similarities and differences.

Reply to comments of Reviewer #3

The authors have thoroughly addressed the reviewers' concerns and comments, and I believe the manuscript is suitable for publication in Nature Communications.

However, I have one additional question. The observed drag phenomenon near charge neutrality is somewhat reminiscent of ambipolar transport of excited carriers in semiconductors. I would appreciate it if the authors could comment on both the similarities and differences.

We thank the Reviewer for the positive evaluation of our work.

To answer the additional question, we have added two paragraphs (Supplementary Note 3E), which discuss both similarities and differences with ambipolar transport in semiconductors and provide key references. We agree that this note would be helpful for readers to navigate the massive literature concerning electron-hole scattering in metals and semiconductors.